# The Effects of Neighborhood Physical and Social Environment on Physical Function among Japanese Community-Dwelling Older Adults: A One-Year Longitudinal Study

**DOI:** 10.3390/ijerph19137999

**Published:** 2022-06-29

**Authors:** Masataka Ando, Naoto Kamide, Miki Sakamoto, Yoshitaka Shiba, Haruhiko Sato, Akie Kawamura, Shuichiro Watanabe

**Affiliations:** 1School of Allied Health Sciences, Kitasato University, 1-15-1 Kitazato, Minami-ku, Sagamihara 252-0373, Japan; naokami@kitasato-u.ac.jp (N.K.); mikis@kitasato-u.ac.jp (M.S.); akie.k@kitasato-u.ac.jp (A.K.); 2Graduate School of Medical Sciences, Kitasato University, 1-15-1 Kitazato, Minami-ku, Sagamihara 252-0373, Japan; 3School of Health Sciences, Fukushima Medical University, 10-6 Sakae-Machi, Fukushima 960-8031, Japan; y-shiba@fmu.ac.jp; 4Faculty of Rehabilitation, Kansai Medical University, 18-89 Uyama Higashi-Machi, Hirakata 573-1136, Japan; satohar@makino.kmu.ac.jp; 5International Graduate School for Advanced Studies, J. F. Oberlin University, 3758 Tokiwa-Machi, Machida 194-0294, Japan; swan@obirin.ac.jp

**Keywords:** community-dwelling older adults, neighborhood environment, physical function, longitudinal study

## Abstract

Previous studies have shown a relationship between physical and social aspects of the neighborhood environment (e.g., built environment, safety) and physical function in older adults. However, these associations are unclear in older Asian adults because longitudinal studies are lacking. This study examined the effects of neighborhood physical and social environment on longitudinal changes in physical function among Japanese older adults. We analyzed 299 Japanese community-dwelling adults aged ≥65 years. Neighborhood environment was assessed using the International Physical Activity Questionnaire Environment Module. Physical function was assessed using handgrip strength, knee extension muscle strength, 5-m walking time, and a timed up-and-go test (TUG) in baseline and follow-up surveys. Changes in physical function over one year were calculated and classified into decline or maintenance groups based on minimal detectable changes. Multiple logistic regression analysis showed that even after adjusting for confounding factors, good access to recreational facilities affected the maintenance of 5-m walking time (odds ratio [OR] = 2.31, 95% confidence interval [CI]: 1.02–5.21) and good crime safety affected the maintenance of TUG (OR = 1.87, 95%CI: 1.06–3.33). Therefore, it is important to assess both physical and social environmental neighborhood resources in predicting decline in physical function among Japanese older adults.

## 1. Introduction

The population is aging all over the world, and Asia has the largest population of older adults of any region [1]. In 2019, the global population aged ≥65 years was 7.7 billion, of which the elderly in Eastern and Southeast Asia, including Japan, accounted for 2.3 billion (30%). This number is estimated to continue increasing until around 2050 [1]. Maintaining physical function in older adults that will enable them to continue to live independently is important in reducing future medical and long-term care costs.

Physical functional decline in older adults is a predictor of the need for long-term care [2,3], disability in activities of daily living (ADL) [4,5], and instrumental ADL [4,6]. Decline in physical function is also associated with falls and fractures [6,7], joint disease [8], dementia [9], and cerebrovascular disease [10], and these diseases and geriatric syndromes are the leading causes of need for long-term care. Physical function is related to diverse factors such as health status [11], physical activity [12], cognitive function [13], and social relationships [14]. More recently, there has been growing interest in the role of environmental factors in maintenance of physical function in older adults. Older adults, in particular, spend increasing amounts of time immersed in the neighborhood where they live as their life space shrinks [15], increasing the likelihood that they will be affected by the environment around their residence; i.e., the neighborhood environment [16].

Although the definition of neighborhood differs among studies, for older adults, neighborhood is defined as the area less than half a mile from home [17] and is considered to be within a walking distance of approximately 10–15 min. The concept of the neighborhood environment includes both physical and social components [18]. The World Health Organization (WHO) advocates the concept of the “age-friendly city,” a physical and social environment that promotes health and participation among older adults [19]. Similarly, a systematic review of studies conducted mainly in the USA and Europe reported that both the physical (e.g., built environment, aesthetics) and social (safety from crime and traffic) environments of the neighborhood are related to physical function for older adults [20]. Thus, evaluation of both these aspects is necessary to determine the effects of the neighborhood environment on physical function.

Given the differences in cultural backgrounds between Western and Asian countries, recent studies conducted in Japan [21,22,23], China [24], and South Korea [25,26,27] have provided perspectives from the Asian region. Of these studies, one has examined the relationship between neighborhood physical and social environments and physical function [26]. However, as most previous studies in Asian countries were cross-sectional studies [21,23,24,25,26,27], it has been difficult to clarify a causal relationship between physical function and neighborhood environment. Our previous study, which evaluated both the physical and social aspects of the neighborhood environment, reported that access to recreational facilities was associated with physical function; however, that too was a cross-sectional study [28].

The International Physical Activity Questionnaire Environmental Module (IPAQ-E) [29] is designed to capture both the physical and the social aspects of the neighborhood environment. This measure enables comprehensive assessment of parameters of the neighborhood environment, such as the built environment and safety, which have been suggested to be related to physical function and can be used to approach the solution of the research question. Therefore, the purpose of this study was to examine the longitudinal effects of neighborhood environment, assessed in terms of both physical and social aspects using the IPAQ-E on physical function.

## 2. Materials and Methods

### 2.1. Study Design and Participants

We conducted a one-year longitudinal study. The participants were recruited from among community-dwelling older adults who participated in health checkups for geriatric syndromes held in Sagamihara City, Kanagawa Prefecture, Japan. Sagamihara City is an ordinance-designated city located in the southwestern part of the Tokyo metropolitan area (population, 726,025; older adults, 26.4%; area, 328.9 km^2^). A detailed description of the procedures and inclusion criteria for this study has been provided elsewhere [28,30]. Briefly, this study included older adults (aged ≥65 years) who were not receiving support under the long-term care insurance system and who had not obtained certification of support or care level. Among 638 new participants in the health check-ups between 2016 and 2018, 14 individuals were excluded because of missing data. Thus, the baseline data were obtained from 624 older adults. In addition, 299 individuals who participated in the follow-up survey one year after the baseline survey and for whom a physical function assessment could be performed were included in the analysis (follow-up rate, 47.9%).

### 2.2. Measurements

#### 2.2.1. Neighborhood Environment

The neighborhood environment was assessed using the Japanese version of the IPAQ-E [31]. The IPAQ-E is a self-administered questionnaire that asks study participants to characterize both the physical and social environments within a 10–15-min walk of their home. The reliability of the IPAQ-E has been verified [31]. Of the total of 17 items in the IPAQ-E, the following 10 items were used in the present study; these meet the definition of neighborhood environment and have been designated as core items and recommended items in previous studies [31,32]. They are: residential density, access to shops, access to public transport (bus stops/stations), presence of sidewalks, presence of bike lanes, access to recreational facilities, crime safety, traffic safety, seeing people being active, and aesthetics. Each item, except for residential density, had a 4-point Likert response scale ranging from strongly disagree to strongly agree. The “residential density” item asks about the main types of houses in the neighborhood (e.g., detached single-family residences, multifamily condos, apartments). The IPAQ-E responses were dichotomized (agree or disagree) and used in the analysis, in accordance with previous studies [31,32]. For residential density, the selection of “detached single-family residences” was categorized as low residential density, and the other selections were categorized as high residential density [31,32]. The detailed questionnaire can be accessed via the following link (https://doi.org/10.1016/j.ypmed.2009.01.014, accessed on 19 May 2022).

#### 2.2.2. Physical Function

As assessments of physical function, muscle strength (handgrip strength and knee extension muscle strength (KEMS)) and physical performance (5-m walking time and timed up-and-go test (TUG) [33]) were measured at two points: at the baseline survey and at the one-year follow-up survey. The participants’ handgrip strength was measured using a Smedley-type dynamometer (T.K.K.5401, TAKEI Scientific Instruments Co., Ltd., Niigata, Japan). The greater of two measurements performed by the dominant hand was adopted as the representative value for handgrip strength. For KEMS, measurements were taken using a handheld dynamometer (µ-Tas F-1; Anima Inc., Tokyo, Japan). The participant sat in a chair with the hip and knee joints in 90° flexion, and isometric knee extension muscle strength at maximum effort was measured in the right leg. The greater of two measurements was adopted as the representative KEMS value. For 5-m walking time, the participants walked at a comfortable pace on a 9-m walkway, consisting of a measurement zone (5 m) and acceleration and deceleration zones (each 2 m); we measured the time taken to walk the 5-m length in the middle of the walkway using a digital stopwatch (ALBA W072; Seiko Watch Corporation, Tokyo, Japan). The faster of two measurements was used as the representative value for the 5-m walking time. For the TUG, the time to stand up from a chair without hand support, walk 3 m as quickly as possible, turn around, walk back, and sit down again [34] was measured with a digital stopwatch (ALBA W072; Seiko Watch Corporation, Tokyo, Japan). The faster of two measurements was used as the representative TUG value.

Changes in each assessed physical function during the year were determined operationally by the following procedure. First, for each participant, the rate of change in each physical function over one year (the amount of change over one year divided by the measured value at baseline) was calculated. A rate of change that declined by more than the minimal detectable change (MDC) was termed “decline,” and a rate of change within the MDC or that improved above the MDC was termed “maintenance.” The MDC represents a boundary value where the amount of change between two repeated measurements is considered to be due to chance fluctuations rather than real change [35]. In the present study, we used MDCs derived from data obtained in a large sample of Japanese community-dwelling older adults [36]. The MDCs for handgrip strength, KEMS, 5-m walking time, and TUG used in this study were 5%, 12%, 7%, and 6%, respectively [36].

#### 2.2.3. Other Variables

The variables of age, sex, height, weight, body mass index (BMI), medical history, pain, medications, habitual exercise, cognitive function, depressive symptoms, social interaction, and functional capacity were also investigated. These variables have been reported in previous studies as factors related to physical function [11,12,13,14]. Medical history (total number of hypertension, diabetes mellitus, dyslipidemia, cerebrovascular disease, and heart disease), pain (low back pain and knee pain), and medications were surveyed using a self-administered questionnaire. Habitual exercise was defined as exercise for 20–30 min or more per session at least two to three times per week. For cognitive function, the Trail Making Test part A (TMT-A) [37] was evaluated. Depressive symptoms were assessed using the five-item version of the Geriatric Depression Scale (GDS-5) [38]. The GDS-5 score ranges 0–5 points, with ≥2 points defined as “with depressive symptoms” [38]. For social interaction, the frequency of interactions per month with non-coresident family and relatives or friends was measured [39]. Functional capacity was assessed by the Tokyo Metropolitan Institute of Gerontology Index of Competence (TMIG-IC) [40]. The TMIG-IC score ranges between 0–13 points, with a higher score indicating greater functional capacity.

### 2.3. Statistical Analysis

For descriptive statistics, continuous variables are presented as means ± standard deviation (SD) and categorical variables as *n* (%). For comparisons between follow-up and non-follow-up subjects and between men and women, we used the unpaired t-test for continuous variables and the chi-square test for categorical variables.

Multiple logistic regression analysis was conducted to examine the effects of the neighborhood environment on physical function, using the change in each physical function over one year (“0” for decline and “1” for maintenance) as the dependent variable and each factor of the neighborhood environment as the independent variable. The adjusted model used age, sex, BMI, each physical function at baseline, habitual exercise, TMT-A, depressive symptoms, and social interaction as confounding factors.

Approximately half of the participants in the baseline survey were able to complete the one-year follow-up survey. Therefore, inverse probability weighting (IPW) methods [41,42] were conducted to test the effect of dropout bias on the relationship between physical function and the neighborhood environment. Propensity scores for the IPW method were calculated by logistic regression analysis with follow-up or non-follow-up as dependent variables and variables that differed by a less than 10% level of statistical significance in comparison between follow-up and non-follow-up subjects as independent variables.

Statistical analysis was performed using IBM SPSS Statistics 27.0 (IBM Japan, Tokyo, Japan), with the level of statistical significance set at 5%.

## 3. Results

### 3.1. Participant Characteristics

Table 1 shows the characteristics of the participants. The mean age was 71.7 ± 4.5 years, 73.9% of participants were women, and the mean TMIG-IC score was 11.9 ± 1.4. Regarding the neighborhood environment, the rates of good access to shops, presence of bike lanes, and good crime safety were significantly higher for men than for women. With regard to physical function, handgrip strength and KEMS were significantly higher in men than in women.

In the comparison between the follow-up and non-follow-up subjects, no significant differences at the 5% level were found for any variable investigated in this study. In contrast, the follow-up subjects tended to have a more extensive medical history (*p* = 0.077, *d* = 0.142), a higher rate of medications (*p* = 0.098, *φ* = 0.066), and a shorter TMT-A (*p* = 0.080, *d* = 0.141) than the non-follow-up subjects. These three variables were used to calculate the propensity scores used in the IPW model described below.

### 3.2. Changes in Physical Function

Table 2 shows the changes in physical function during the one-year follow-up. The percentage of participants with a decline in each function ranged from 13.7% (5-m walking time) to 34.8% (handgrip strength). The percentage of women with declined KEMS was significantly higher than that of men.

### 3.3. Neighborhood Environment and Physical Function

Table 3 shows the effects of neighborhood environment factors on changes in physical function over one year. Multiple logistic regression analysis showed that even after adjusting for confounding factors, good access to recreational facilities affected the maintenance of 5-m walking time (OR = 2.31, 95%CI: 1.02–5.21) and good crime safety affected the maintenance of TUG (OR = 1.87, 95%CI: 1.06–3.33).

To consider the influence of dropout bias on the results, IPW was conducted. After IPW, the effect of access to recreational facilities on 5-m walking time (OR = 2.31, 95%CI: 1.01–5.27) and the effect of crime safety on TUG (OR = 1.94, 95%CI: 1.10–3.43) were still significant.

## 4. Discussion

In this study, the effect of the neighborhood environment, assessed in terms of both physical and social aspects, on changes in physical function over a one-year period was examined using multiple logistic regression analysis. Previous studies in Western countries have shown that physical environment [43,44] and social environment [45,46] are longitudinally related to physical function. In comparison, in Asian countries, we could find only one longitudinal study that investigated the effect of the neighborhood environment on the physical function of older Japanese adults [22]. This previous study evaluated only the neighborhood physical environment; furthermore, assessment of physical function was limited to handgrip strength. Therefore, the present study is the first longitudinal study in the Asian region to examine the effects of both physical and social aspects of the neighborhood environment on physical function, including muscle strength (handgrip strength and KEMS) and physical performance (5-m walking time and TUG).

The results of multiple logistic regression analysis showed that older adults living in environments with good access to recreational facilities maintained their 5-m walking time performance after one year. Although positive associations between physical function and recreational facilities, which are factors of the physical environment, have been reported in cross-sectional studies [23,26], the present results strengthen the findings because of the longitudinal study design. In addition, good crime safety influenced the maintenance of TUG performance after one year. Previous studies in the USA have shown that the risk of subjectively assessed mobility limitation increases with neighborhood disorders such as crime and vandalism [47]. The results of the present study show that in the Asian context, crime safety, a factor of the social environment, has an impact on objectively assessed changes in physical function. Thus, both the physical and social environments affect physical function, especially physical performance, in older Japanese adults.

There are two possible mechanisms in the relationship between recreational facilities and 5-m walking time. The first pathway is the increase in physical activity associated with walking. It has been reported that older adults who have good access to recreational facilities undertake more total walking in the neighborhood per week [32]. Another pathway is the promotion of health literacy. It is assumed that older adults who have recreational facilities nearby have more opportunities to interact with fellow facility users and to participate in health-related events at the facilities. It is considered that they would acquire health knowledge and improve their health management skills accordingly. In older adults, health literacy has also been reported to be associated with access to health care (e.g., primary care, preventive services) [48]. The 5-m walk time is an indicator of physical performance as well as a reflection of overall health status [49] and may be influenced by the protective effects of health literacy. In contrast, the possible mechanisms for the relationship between crime safety and TUG include increased social activity. Safety in neighborhoods affects people’s health as a social relationship [50]. It has also been reported that social isolation is associated with future decline of TUG performance [42]. In social activities, older adults are required to perform more advanced physical tasks than simple walking, such as vertical shifts in the center of gravity and changes in direction, depending on the situation of the individual and the objects in the social activity setting, and these may have had a specific effect on TUG. Further study is needed to verify these mechanisms.

This study has several limitations. First, the follow-up rate of the participants in this study was about one half. In this regard, we confirmed that there was little difference in the attributes of the two groups in comparison between the follow-up and non-follow-up groups and that the results of the multiple logistic regression analysis remained unchanged after implementation of the IPW method. In other words, we considered that dropout bias had a very minor effect on the results of this study. However, these validations do not completely eliminate the influence of dropout bias on the results. Second, the one-year follow-up period of this study is short. By defining decline or maintenance of physical function based on MDC, we detected changes in physical function that were not at least within the range of measurement error. However, MDC captured only minimal changes in physical function, and it is unclear whether the neighborhood environment has an effect on clinically meaningful changes in physical function. We cannot rule out the possibility that the results could change if a different MDC was used. In addition, this study evaluated the neighborhood environment in a subjective manner. The participants were older adults with a rather high functional capacity, and we consider that the quality of subjective evaluation of the neighborhood environment was assured; however, it cannot be denied that different results could possibly be obtained if an objective index is used. At the very least, the parameters used in this study, such as residential density, access to destinations, the presence of the neighborhood’s infrastructure, and the crime rate, can be objectified by geographic information systems. It is necessary to verify whether the same results would be obtained when using objective indicators of the neighborhood environment. Finally, this study did not examine the years of residence in the neighborhood or socioeconomic status, and adjustment for these factors was inadequate.

## 5. Conclusions

The results of this study suggest that good access to recreational facilities and good crime safety affect the maintenance of physical function after one year among older Japanese adults living in the community. Therefore, it is important to evaluate both the physical and social environments in the neighborhood to predict short-term decline in physical function. Further research is needed on the long-term effects of the neighborhood environment on physical function in older adults in Asian countries, which have an increasingly aging population.

## Figures and Tables

**Table 1 ijerph-19-07999-t001:** Participant characteristics at baseline.

Variable	Overall	Men	Women	*p*-Value *
	*n* = 299	*n* = 78	*n* = 221	
Age, y	71.7 ± 4.5	73.4 ± 4.8	71.1 ± 4.3	<0.001
Sex, women	221 (73.9)			
BMI, kg/m^2^	22.0 ± 3.1	22.4 ± 2.2	21.9 ± 3.3	0.035
No. of medical history items	0.94 ± 0.9	1.18 ± 1.0	0.86 ± 0.8	0.006
Low back pain, yes	109 (36.5)	31 (39.7)	78 (35.3)	0.483
Knee pain, yes	106 (35.5)	20 (25.6)	86 (38.9)	0.035
Medications, yes	210 (70.2)	59 (75.6)	151 (68.3)	0.224
Habitual exercise, yes	229 (76.6)	61 (78.2)	168 (76.0)	0.695
TMT-A, s	56.2 ± 21.5	61.4 ± 34.3	54.4 ± 14.2	0.065
Depressive symptoms, yes	45 (15.1)	5 (6.4)	40 (18.1)	0.013
Social interaction, times/month	23.6 ± 16.9	19.2 ± 16.7	25.1 ± 16.7	0.008
TMIG-IC, /13 points	11.9 ± 1.4	11.7 ± 1.7	12.0 ± 1.3	0.092
Neighbor environment				
Residential density, high	79 (26.4)	23 (29.5)	56 (25.3)	0.475
Access to shops, good	224 (74.9)	68 (87.2)	156 (70.6)	0.004
Access to public transport, good	274 (91.6)	74 (94.9)	200 (90.5)	0.230
Presence of sidewalks, yes	210 (70.2)	54 (69.2)	156 (70.6)	0.822
Presence of bike lanes, yes	117 (39.1)	40 (51.3)	77 (34.8)	0.011
Access to recreational facilities, good	252 (84.3)	69 (88.5)	183 (82.8)	0.238
Crime safety, good	192 (64.2)	60 (76.8)	132 (59.7)	0.006
Traffic safety, good	204 (68.2)	53 (67.9)	151 (68.3)	0.951
Seeing people being active, yes	253 (84.6)	68 (87.2)	185 (83.7)	0.465
Aesthetics, good	210 (70.2)	56 (71.8)	154 (69.7)	0.726
Physical function				
Handgrip strength, kgf	26.7 ± 6.5	34.8 ± 6.1	23.8 ± 3.6	<0.001
KEMS, kgf	29.3 ± 9.2	34.8 ± 10.4	27.3 ± 7.8	<0.001
5-m walking time, s	3.4 ± 0.5	3.4 ± 0.5	3.4 ± 0.4	0.233
TUG, s	5.7 ± 0.9	5.6 ± 1.0	5.8 ± 0.8	0.164

Note. Values are mean ± SD or *n* (%). * *p*-value for comparison between sex (unpaired t-test or chi-square test). Abbreviation: SD, standard deviation; BMI, body mass index; TMT-A, Trail Making Test, part A; TMIG-IC, Tokyo Metropolitan Institute of Gerontology Index of Competence; KEMS, knee extension muscle strength; TUG, timed up-and-go test.

**Table 2 ijerph-19-07999-t002:** Changes in physical function during one year of follow-up.

Variable	Overall	Men	Women	*p*-Value *
	*n* = 299	*n* = 78	*n* = 221	
Handgrip strength (≥5% decline)	104 (34.8)	32 (41.0)	72 (32.6)	0.178
KEMS (≥12% decline)	102 (34.2)	17 (21.8)	85 (38.6)	0.007
5-m walking time (≥7% decline)	41 (13.7)	11 (14.1)	30 (13.6)	0.907
TUG (≥6% decline)	69 (23.1)	14 (17.9)	55 (24.9)	0.211

Note. Values are *n* (%). * *p*-value for comparison between sex (chi-square test). Abbreviation: KEMS, knee extension muscle strength; TUG, timed up-and-go test.

**Table 3 ijerph-19-07999-t003:** Effects of perceived neighborhood environment on physical function by multiple logistic regression analysis.

	Handgrip Strength		KEMS		5-m Walking Time		TUG	
	OR	95%CI	OR	95%CI	OR	95%CI	OR	95%CI
Residential density (ref: low)								
Unadjusted model	0.83	0.49–1.41	0.84	0.49–1.44	1.32	0.60–2.91	0.93	0.51–1.70
Adjusted model	0.84	0.49–1.46	0.78	0.45–1.37	1.24	0.55–2.77	1.02	0.54–1.91
IPW model	0.85	0.49–1.49	0.76	0.44–1.32	1.19	0.52–2.74	1.06	0.57–1.99
Access to shops (ref: poor)								
Unadjusted model	0.61	0.34–1.08 †	0.74	0.42–1.31	0.82	0.37–1.80	1.43	0.79–2.59
Adjusted model	0.66	0.37–1.20	0.62	0.35–1.12	0.83	0.37–1.86	1.41	0.76–2.64
IPW model	0.67	0.37–1.21	0.60	0.33–1.09 †	0.82	0.37–1.82	1.52	0.81–2.85
Access to public transport (ref: poor)								
Unadjusted model	0.89	0.45–2.49	0.90	0.37–2.15	0.85	0.24–2.97	1.06	0.41–2.76
Adjusted model	1.10	0.46–2.64	0.78	0.32–1.91	0.84	0.23–3.00	1.12	0.41–3.02
IPW model	1.12	0.46–2.72	0.73	0.31–1.72	0.86	0.23–3.13	1.25	0.46–3.40
Presence of sidewalks (ref: no)								
Unadjusted model	1.32	0.79–2.21	0.67	0.39–1.15	1.11	0.55–2.26	1.60	0.91–2.83
Adjusted model	1.32	0.78–2.23	0.67	0.39–1.17	1.12	0.54–2.32	1.58	0.88–2.83
IPW model	1.28	0.75–2.19	0.68	0.39–1.18	1.05	0.51–2.19	1.58	0.87–2.87
Presence of bike lanes (ref: no)								
Unadjusted model	0.82	0.50–1.33	1.19	0.72–1.95	1.01	0.51–1.98	1.50	0.85–2.66
Adjusted model	0.88	0.54–1.46	1.08	0.65–1.81	0.99	0.49–1.98	1.46	0.81–2.66
IPW model	0.88	0.53–1.45	1.09	0.65–1.83	1.00	0.51–1.98	1.51	0.82–2.78
Access to recreational facilities (ref: poor)								
Unadjusted model	1.20	0.63–2.28	0.61	0.30–1.24	2.26	1.04–4.91 *	1.52	0.76–3.04
Adjusted model	1.30	0.67–2.53	0.52	0.25–1.09 †	2.31	1.02–5.21 *	1.61	0.78–3.33
IPW model	1.30	0.67–2.52	0.55	0.27–1.12	2.31	1.01–5.27 *	1.62	0.80–3.31
Crime safety (ref: poor)								
Unadjusted model	1.44	0.88–2.36	0.94	0.60–1.62	1.18	0.60–2.31	1.93	1.12–3.34 *
Adjusted model	1.45	0.87–2.41	0.91	0.54–1.53	1.17	0.58–2.37	1.87	1.06–3.33 *
IPW model	1.38	0.83–2.30	0.92	0.54–1.58	1.14	0.58–2.24	1.94	1.10–3.43 *
Traffic safety (ref: poor)								
Unadjusted model	0.66	0.39–1.11	0.92	0.55–1.55	0.76	0.36–1.59	1.30	0.74–2.29
Adjusted model	0.63	0.37–1.08 †	0.96	0.56–1.64	0.74	0.35–1.58	1.22	0.68–2.20
IPW model	0.61	0.36–1.05 †	0.90	0.52–1.56	0.72	0.34–1.51	1.21	0.67–2.17
Seeing people being active (ref: no)								
Unadjusted model	1.00	0.52–1.93	1.03	0.53–1.99	0.56	0.19–1.64	1.58	0.79–3.16
Adjusted model	0.99	0.50–1.94	0.98	0.50–1.94	0.59	0.20–1.78	1.70	0.83–3.50
IPW model	0.95	0.48–1.90	0.98	0.49–1.96	0.55	0.18–1.66	1.66	0.81–3.42
Aesthetics (ref: poor)								
Unadjusted model	1.32	0.79–2.21	1.20	0.71–2.01	1.11	0.55–2.26	1.04	0.58–1.87
Adjusted model	1.40	0.82–2.37	1.28	0.75–2.19	1.09	0.52–2.27	1.06	0.58–1.93
IPW model	1.35	0.78–2.31	1.22	0.71–2.09	1.05	0.50–2.20	1.01	0.55–1.86

Note. Dependent variables: changes in each physical function over one year (0 = decline, 1 = maintenance). Independent variables: each factor of the neighborhood environment. Adjusted model and IPW model: adjusted for age, sex, BMI, each physical function (at baseline), habitual exercise, TMT-A, depressive symptoms, and social interaction. * *p* < 0.05, † *p* < 0.10. Abbreviation: KEMS, knee extension muscle strength; TUG, timed up-and-go test; OR, odds ratio; CI, confidence interval; IPW, inverse probability weighting; BMI, body mass index; TMT-A, Trail Making Test part A.

## Data Availability

Data can be provided on request from the corresponding authors.

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
