# Peer review of "The Effects of Neighborhood Physical and Social Environment on Physical Function among Japanese Community-Dwelling Older Adults: A One-Year Longitudinal Study"

_ijerph, 2022, doi:10.3390/ijerph19137999_

Round 1

Reviewer 1 Report

Comments

1.     The research assesses the Effects of Neighborhood Environment on Physical Function among Japanese Community-Dwelling Older Adults. One of the major issues of the work is the term neighborhood environment is not defined accurately in the beginning of the study.

2.     There are different factors that defines a neighborhood environment, in terms of socio economic context, environmental context or geographical/physical context etc. By looking at the title or abstract, it is difficult to understand the context the author Is referring to and as well as the variables of the neighborhood environment. This needs to be clarified.

3.     Table number 3 – variable selection, is there a specific method followed? What is the basis for selecting these variables?

4.     It is better to explain the questions and parameters that are being focused on the “International Physical Activity 24 Questionnaire Environment Module”. Better to explain this survey in the introduction.

Reviewer 2 Report

Line 83 – please explain what population aging rate is and how it is calculated. Or, if it simply means percent of older adults, then please change this term and define “older adults”

Lines 99-108 -  It is absolutely necessary to include a table with questions from the Questionnaire (make a table similar to Table 1 from reference [28] - https://www.semanticscholar.org/paper/Association-of-physical-activity-and-neighborhood-Inoue-Murase/f8c82b791995aae57e1b07502c3ad866ebdcd119/figure/0) The reader should have easy access to this information within this article.  I had to search for reference  [28] online to find this questionnaire.

Line 108 – explain how responses to residential density question were converted into a dichotomous variable

Line 140 – explain how these values (5, 12, 7 and 6%) were chosen as MDC values. The abstract for reference [33] does not explain it

Discussion – it is important to discuss the importance of selected thresholds for MDC. These seem somewhat arbitrary, so it is a limitation of the study. If you chose different values for MDC – how would your results look? Consider doing a sensitivity analysis

Line 297 – an objective index. Please discuss how other research methods, such as GIS-derived (geographic information systems) measures of access, could be used here. At least six of your variables (three access-related, two presence-related, and residential density) could be calculated in GIS using available  GIS data for the city.  These would provide an objective measure of the built environment.
